# Emergence of *Anaplasma* Species Related to *A. phagocytophilum* and *A. platys* in Senegal

**DOI:** 10.3390/ijms24010035

**Published:** 2022-12-20

**Authors:** Rosanna Zobba, Claudio Murgia, Mustapha Dahmani, Oleg Mediannikov, Bernard Davoust, Roberta Piredda, Eleonora Schianchi, Alessandra Scagliarini, Marco Pittau, Alberto Alberti

**Affiliations:** 1Dipartimento di Medicina Veterinaria, University of Sassari, 07100 Sassari, Italy; 2IRD, AP-HM, MEPHI, Aix Marseille University, 13001 Marseille, France; 3IHU Méditerranée Infection, 13005 Marseille, France; 4Dipartimento di Medicina Specialistica, Diagnostica e Sperimentale, University of Bologna, 40126 Bologna, Italy

**Keywords:** obligate intracellular bacteria, *Anaplasma* diversity, tick-borne diseases, zoonosis, one health

## Abstract

The genus *Anaplasma* (*Anaplasmataceae, Rickettsiales*) includes tick-transmitted bacterial species of importance to both veterinary and human medicine. Apart from the traditionally recognized six Anaplasma species (*A. phagocytophilum*, *A. platys*, *A. bovis*, *A. ovis, A. centrale*, *A. marginale*), novel strains and candidate species, also of relevance to veterinary and human medicine, are emerging worldwide. Although species related to the zoonotic *A. platys* and *A. phagocytophilum* have been reported in several African and European Mediterranean countries, data on the presence of these species in sub-Saharan countries are still lacking. This manuscript reports the investigation of *Anaplasma* strains related to zoonotic species in ruminants in Senegal by combining different molecular tests and phylogenetic approaches. The results demonstrated a recent introduction of *Candidatus* (*Ca*) Anaplasma turritanum, a species related to the pathogenic *A. platys*, possibly originating by founder effect. Further, novel undetected strains related to *Candidatus* (*Ca*) Anaplasma cinensis were detected in cattle. Based on *groEL* and *gltA* molecular comparisons, we propose including these latter strains into the *Candidatus* (*Ca*) Anaplasma africanum species. Finally, we also report the emergence of *Candidatus* (*Ca*) *A. boleense* in Senegal. Collectively, results confirm that *Anaplasma* species diversity is greater than expected and should be further investigated, and that *Anaplasma* routine diagnostic procedures and epidemiological surveillance should take into account specificity issues raised by the presence of these novel strains, suggesting the use of a One Health approach for the management of *Anaplasmataceae* in sub-Saharan Africa.

## 1. Introduction

Bacteria belonging to *Anaplasmataceae* (*Alphaproteobacteria*: *Rickettsiales*) are tick-transmitted, Gram-negative, obligate intracellular bacteria that replicate in both vertebrate and invertebrate host cells [1] and are significantly relevant both to veterinary and public health [2]. According to Dumler and coworkers [1], this family comprises four classified genera (*Anaplasma*, *Ehrlichia*, *Neorickettsia*, and *Wolbachia*), with the genus *Anaplasma* currently including six species with variable pathogenicity [1,2,3]. Additionally, novel strains and candidate species [2,4] have been recently recorded (Table 1).

*A. phagocytophilum* is the most relevant species in terms of animal and human tick-borne diseases within the genus, being the causative agent of ruminant tick-borne fever and granulocytic anaplasmosis of horses, dogs, and humans. Similarly, *A. platys* and *A. marginale* are of importance in veterinary medicine, as they cause cyclic thrombocytopenia in dogs and bovine anaplasmosis, respectively.

In addition to *A. phagocytophilum*, *A. platys*, *A. bovis*, *A. capra,* and *A. ovis* have been shown to be pathogenic to humans (Table 1).

In the last 10 years, novel *Anaplasma* strains related to but genetically distinct from the zoonotic *A. phagocytophilum* and *A. platys* have been identified worldwide. 

In Japan, bacteria phylogenetically clustering in a monophyletic clade distinct from but closely related to *A. phagocytophilum* have been identified in *Haemaphysalis* and *Ixodes* ticks infesting domestic and wild ruminants [17,21], and in China, *Anaplasma* strains designated as *Candidatus (Ca)* Anaplasma boleense were detected in *Hyalomma asiaticum* ticks infesting domestic ruminants [20]. Both the Japanese strains and *Ca* A. boleense have been identified in the Mediterranean area (Tunisia, Italy), where they were respectively described as *A. phagocytophilum*-like 2 and *A. phagocytophilum*-like 1, based on their *16S rRNA* and *groEL* gene sequences [3,22,23]

Based on *16S rRNA*, *groEL*, and *gltA*, *Anaplasma* strains related to the canine *A. platys* have been identified in Sardinian ruminants, cats, and *Rhipicephalus* ticks. Based on genetic comparisons, these strains were assigned to *A. platys*-like [2,10,11,24]. The same organism was detected in Tunisian ruminants [25] and in ticks from Costa Rica [26].

In China, novel *Anaplasma* strains genetically related to *A. platys* were identified in *Rhipicephalus microplus* by Guo and colleagues [12]. Recently, upon *groEl* and *gltA* comparisons, Mediterranean and Chinese *A. platys*-like strains were respectively assigned to the two novel, distinct species *Candidatus* (*Ca)* Anaplasma turritanum and *Ca* Anaplasma cinensis [2].

Reports on *A. phagocytophilum*, *A. platys,* and related strains are scarce in sub-Saharan Africa. Notably, Dahmani and colleagues [27] recorded the emergence of a potentially new species commonly infecting ruminants in Senegal, and provisionally named it *Anaplasma* cf. *platys* by comparison of concatenated *23S rRNA*, *16S rRNA,* and *rpoB* genes.

This paper investigates the presence of *Anaplasma* strains related to the zoonotic *A. phagocytophilum* and *A. platys* in Senegal ruminants by combining molecular tools targeting the *Anaplasma 16S rRNA*, *groEL,* and *gltA* genes. Sequencing, molecular typing, and phylogenetic analyses allowed us to demonstrate the emergence of *Candidatus* A. turritanum in Senegal and to postulate its recent introduction in the Mediterranean area by the founder effect. Moreover, the presence of *A. phagocytophilum*-like 2 and of novel bovine *Anaplasma* strains related to *Ca* A. cinensis is demonstrated for the first time in Senegal. Implications of the emergence of these *Anaplasma* species in sub-Saharan Africa on diagnostics, diversity, transmission, and public health are also discussed.

## 2. Results

The use of three distinct PCR tests for the detection of Anaplasma strains related to *A. platys*, combined with sequencing and Blast analysis (Table 2 and Table 3), allowed us to establish the presence of *Ca* A. turritanum in 92/176 ruminants (52%). Of these, 83/176 were sheep (47%), and 9/176 (0.05%) were goats, whereas *Ca* A. turritanum was not detected in bovines. The number of animals testing positive by *groEL* and *gltA* PCR were comparable. Sequencing of *groEL* amplicons obtained from 37 sheep and six goats revealed the presence of a unique sequence type with 98–100% similarity/homology to *Ca* A. turritanum strains isolated from cats and ruminants in Tunisia [25] and in Italy [10,11]. Similarly, *gltA* sequencing from 34 sheep and seven goats revealed the presence of a unique sequence type that was 100% homologous to the same ruminant strains.

Novel Anaplasma strains (Table 2 and Table 3) were also detected in three varieties of cattle by both *groEL* and *gltA* PCRs. *GroEL* sequences were assigned to three distinct sequence types, whereas *gltA* sequencing generated an invariable sequence for the three animals and therefore a unique sequence type. *GroEL* sequence types showed a higher level of similarity (Table 3) with *Anaplasma* spp. isolated in Tunisian cattle [25] and droedaries [28] and with *Ca* A. cinensis strains isolated in China [12]. The invariable *gltA* sequence detected in the three animals was consistently more similar (Table 3) to Anaplasma spp. strains isolated in dromedaries in Egypt [29]) and to *Ca* A. cinensis isolated in China [12].

One ovine sample was unexpectedly positive for groEL semi-nested PCR specific for *A. phagocytophilum* (Table 3) and resulted in 88% similarity to various *A. bovis* strains.

In phylogenetic trees (Figure 1), consistentl with BLASTN comparisons, the groEL invariable sequence type Ovicaprine1 Senegal is included in a statistically supported monophyletic clade together with *Ca* A. turritanum *groEl* sequences isolated in ticks and ruminants in Tunisia, Italy, and Costa Rica. This clade is closely related to *A. platys* and is basal to *Ca* A. cinensis. Interestingly, the three *groEL* sequence types Bovine1, Bovine2, and Bovine3 were grouped together in a statistically supported clade, together with *groEL* sequences isolated in dromedaries and cattle from Tunisia; this clade formed a Ca A. cinensis sister group.

*GltA* phylogeny confirms that which was observed by *groEL* (Figure 2). The sequence type Ovicaprine2 Senegal groups with *Ca* A. turritanum sequences obtained from ruminants in Tunisia and Italy confirmed the circulation of this Anaplasma species in Senegal. Further, the sequence type Bovine5 Senegal, similar to that observed with *groEL* sequence types obtained from the same animals, is included together with *gltA* sequences rescued from dromedaries in Egypt in a statistically supported group distinct from but related to *Ca* A. cinensis.

Investigation of strains related to *A. phagocytophilum* by nested *16S rRNA* PCR resulted in 28/176 positive samples (23 sheep and five cattle, Table 2). Sequencing of *16S rRNA* PCR products allowed us to assign sequences to four sequence types (Ovibovine2 Senegal, Ovine3 Senegal, Ovine4 Senegal, and Ovine5 Senegal) mostly similar to *Ca* A. boleense (*A. phagocytophilum*-like2. Table 2). Consistent with *16S rRNA* results, out of the three PCR tests targeting the groEL gene, only the semi-nested PCR targeting the *A. phagocytophilum*-like2 (*Ca* A. boleense) *groEL* gene showed positivity (Table 2). Out of 28 *16S rRNA* PCR-positive animals, 23 were also positive for *groEL* (18 sheep and five cattle). Sequencing revealed the circulation of three sequence types; a sequence type rescued exclusively from cattle (Bovine4 Senegal), a sequence type circulating only in sheep (Ovine1 Senegal), and finally a sequence type (Ovibovine1 Senegal) rescued from both cattle and sheep.

GroEL phylogeny of the three sequence types (Figure 3), together with sequences representative of the different Anaplasma strains, placed Ovine1 Senegal, Bovine4 Senegal, and Ovibovine Senegal in a monophyletic clade, including *Ca* A. boleense sequences isolated in Chinese ticks, consistent with *16S rRNA* and *groEL* BLASTN results.

## 3. Discussion

Apart from the six widely recognized species included in the genus *Anaplasma*, novel strains and candidate species have been reported in the last 10 years worldwide. The emergence of these novel strains related to *A. phagocytophilum* and *A. platys*, together with the acknowledgment of *A. platys* and *A. capra* as zoonotic agents, has raised concerns in both veterinary and public health.

Among species related to *A. platys*, *Ca* A. turritanum was recently identified and described in several European and African Mediterranean Countries [2]. Reports of these strains in African sub-Saharan countries are still lacking.

This study reports the emergence of *Ca* A. turritanum in a sub-Saharan African country (Senegal). By considering cumulative positivity to both *groEL* and *gltA* PCR, *Ca* A. turritanum was detected in some 50% of the tested ruminants from Senegal, with a prevalence consistent with values previously reported in the Mediterranean area [10]. *Ca* A. turritanum prevalence seems higher in sheep (83/134, 62%) with respect to goats (9/28, 32%), although it should be taken into consideration that the number of sampled animals was much lower in goats than in sheep. Further, *Ca* A. turritanum was not detected in bovines. Cohen’s kappa coefficient (κ) was 0.97, indicating almost perfect agreement between *Ca* A. turritanum *groEL* and *gltA* pcr tests and validating their use in the diagnostics routine.

Considering *groEL* and *gltA* results obtained in this study, *Anaplasma* cf. platys strains previously identified in Senegal by Dahamani and coworkers [27] by comparing concatenated 23S rRNA, 16S rRNA, and the rpoB genes in the same animals should be assigned to the *Ca* A. turritanum species.

In spite of a high prevalence of *Ca* A. turritanum in Senegal, ruminants of both *gltA* and *groEL* sequencing resulted in a single sequence type circulating in sheep and goats from Senegal; taking into account the high genetic variability of *Ca* A. turritanum in Italian and Tunisian ruminants [2], we postulate that *Ca* A. turritanum originated in the Mediterranean area, and that its expansion to Sub-Saharan Countries was hampered by the Sahara Desert, which worked as a natural barrier to hosts’ and vectors’ diffusion and contact. Under this scenario, the absence of genetic variability of *Ca* A. turritanum in Senegal could be explained by its recent introduction and by the founder effect possibly resulting from the ingress of a single *Ca* A. turritanum strain in this country, for instance, through zootechnical practices or carried by ticks transported by migratory birds.

Furthermore, based on *gltA* and *groEL* comparisons (Figure 1 and Figure 2), the emergence of novel *Anaplasma* strains forming a monophyletic clade closely related to the recently described *Ca* A. cinensis [2,12] is also reported in Senegalese cattle. Considering *groEL* and *gltA* philogenies, we propose assigning these latter novel strains to the novel species *Ca* A. africanum. The *Ca* A. africanum clade includes strains previously described in dromedary camels from Egypt and reported as *Anaplasma* sp. [29], in dromedaries from Tunisia, deposited in the GenBank under *A. platys*-like designation [28], and in cattle from Tunisia [25]. Finally, the emergence of strains belonging to *Ca* A. boleeense is demonstrated in Senegal by the identification of three *groEL* sequence types in ruminants (Table 3), included in a monophyletic cluster together with strains rescued from ticks in China (Figure 3).

The identification of new *Anaplasma* strains related to *A. platys* and *A. phagocytophilum* (pathogenic to both animals and humans) raises concerns in the management of these infections in sub-Saharan Africa and points to the importance of a One Health approach in taking actions in order to establish geographical distribution, host and vector tropism, and pathogenicity of these novel strains. Further, for some of them, the acknowledgment of their zoonotic potential (e.g., *A. capra*) reinforces the hypothesis that *Anaplasma* diversity and the number of potential zoonotic species included in this genus could be greater than that suspected in the past, and it could justify more effort in investigating the presence of possible emerging novel strains worldwide.

In conclusion, we recommend that these novel strains should be included in the diagnostic routine and epidemiological surveillance of tick-transmitted pathogens. Finally, serological and molecular diagnostic tools and past data should be reconsidered in light of the possibility of coinfection with traditional *Anaplasma* species, routinely diagnosed in the past, and these novel genetically related strains.

## 4. Materials and Methods

### 4.1. Ethics Approval and Consent to Participate

Animal blood samples were collected by veterinarians according to good practice and following Senegalese regulations with the agreement of owners.

### 4.2. Samples Collection and DNA Extractions

A total of 176 EDTA blood samples were used in this study. Blood was collected from 176 clinically healthy ruminants (Table 2) in June 2014 in the Keur Momar Sarr Senegal region (15°55′0.0012″ N, 15°58′0.0012″ W). Samples were stored at −20 °C until use. After thawing (before DNA extraction), blood samples were split into 100µL aliquots. DNA was extracted from 100 µL blood aliquots with the DNeasy Blood and Tissue Kit (Qiagen, Milano, Italy) according to vendor instructions.

### 4.3. PCR Strategies

The presence of species related to *A. phagocytophilum* and *A. platys* in samples was investigated using different PCR tests targeting the *16S rRNA* and *groEL* and *gltA* genes (Table 2 and Table 3). To investigate strains related to *A. platys,* 3 different PCR tests were used: (1) a semi-nested PCR targeting 515 bp of the *Ca* A. turritanum *groeEL* gene [30,31]; (2) a semi-nested PCR targeting 947 bp of *Ca* A. turritanum *gltA* gene [2]; (3) a semi-nested PCR targeting 660 bp of the *gltA* gene of species related *A. platys* [12]. To investigate strains related to *A. phagocytophilum,* samples were initially screened with a nested PCR test [22,23] specific to the *16S rRNA* gene of the *A. phagocytophilum* group (*A. phagocytophilum*, *A. phagocytophilum*-like1 (Japanese strains)), *A. phagocytophilum*-like2 (*Ca* A. boleense). Samples positive for *16S rRNA PCR* were screened with 3 additional PCR tests (Table 3): (1) a semi-nested PCR targeting 573 bp of the *A. phagocytophilum groEL* gene [30,31]; (2) a nested PCR targeting 1446 bp of the *A. phagocytophilum*-like1 (Japanese strains) *groEL* gene [17]; (3) a semi-nested PCR targeting 792 bp of the *A. phagocytophilum*-like2 (*Ca* A. boleense) *groEL* gene [3].

For all PCR tests, cycling conditions and mixing were as described in the original papers.

### 4.4. Sequencing, Sequence Types Assignment, and Phylogenetic Analyses

Amplicons were purified by using the DNA Clean and Concentrator kit (Zymo Research, Milano, Italy), according to the manufacturer’s instruction. DNA samples’ quality and quantity were assessed using the Nano Drop (Eppendorf, Milano, Italy). PCR products were automatically sequenced (BMR Genomics, Padova, Italy) on both strands. Chromatograms were edited with Chromas 2.2 (Technelysium, Helensvale, Australia), and sequences were aligned with Clustal X version 2.0 [32]. All sequences obtained in this study were deposited in the GenBank database (accession numbers are shown in Table 3).

Sequences were assigned to unique sequence types named after the host and identified by progressive numbers. Sequence types were challenged against the GenBank database with standard nucleotide BLAST (BlastN; https://blast.ncbi.nlm.nih.gov, accessed on 1 September 2022). Sequence types were also used as operational taxonomic units (OTUs) in phylogenetic analyses.

In particular, groEL sequence types obtained in this study by 2 semi-nested PCRs (Table 3) were aligned with 48 unique *groEL* sequences representative of *Anaplasma* species identified in ticks and vertebrate hosts worldwide. Alignment was inputted in MEGA11 [33] to reconstruct OTUs phylogeny. Phylogenetic analysis was inferred using the maximum likelihood method based on the Tamura 3-parameter model [34], identified as the best model by MEGA11. The robustness of trees was evaluated by bootstrapping over 1000 reiterations [35]. Initial tree(s) for the heuristic search were obtained automatically by applying Neighbor-Join and BioNJ algorithms to a matrix of pairwise distances estimated using the Tamura 3-parameter model and then selecting the topology with a superior log likelihood value. A discrete Gamma distribution was used to model evolutionary rate differences among sites (5 categories (+G, parameter = 1.3501)). This analysis involved 53 nucleotide sequences. There were 405 positions in the final dataset.

Furthermore, gltA sequence types obtained in this study by 2 semi-nested PCRs (Table 3) were aligned with 42 unique *gltA* sequences representative of *Anaplasma* species identified in ticks and vertebrate hosts worldwide. Alignment was inputted in MEGA11. Phylogeny was reconstructed using the maximum likelihood method based on the Tamura 3-parameter model and evaluated by bootstrapping over 1000 replicates. Initial tree(s) for the heuristic search were obtained automatically by applying Neighbor-Join and BioNJ algorithms to a matrix of pairwise distances estimated using the Tamura 3-parameter model and then selecting the topology with a superior log likelihood value. A discrete Gamma distribution was used to model evolutionary rate differences among sites (5 categories (+G, parameter = 2.3476)). This analysis involved 44 nucleotide sequences. There were 483 positions in the final dataset.

Finally, *groEL* sequence types obtained in this study by semi-nested PCR (Table 3) were aligned to 26 unique *groEL* sequences representative of *Anaplasma* species identified in ticks and vertebrate hosts worldwide. As above, trees were reconstructed using the maximum likelihood method based on the Tamura 3-parameter model and evaluated by bootstrapping. In this case, a discrete Gamma distribution was used to model evolutionary rate differences among sites (5 categories (+G, parameter = 0.5603)). The analysis involved 29 nucleotide sequences, and there were 667 positions in the final dataset.

## Figures and Tables

**Figure 1 ijms-24-00035-f001:**
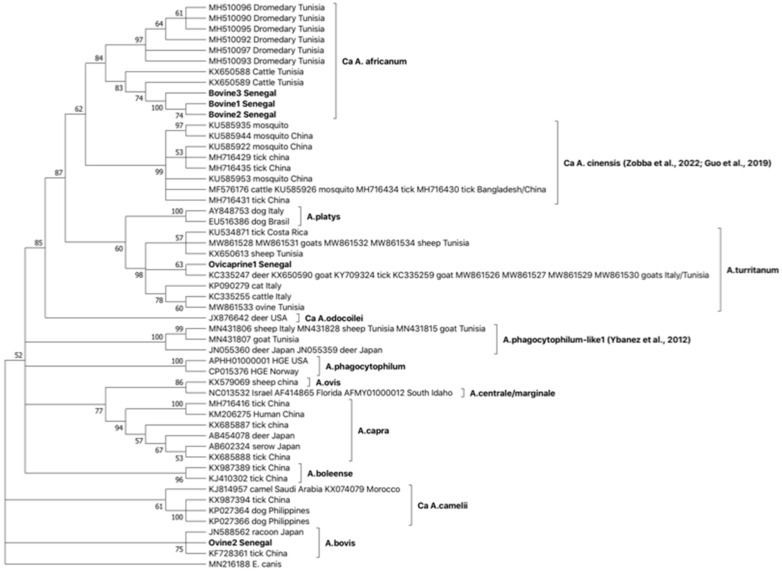
Phylogeny of the 4 *groEL* sequence types identified in Senegal and genetically related to *A. platys* with OTUs selected as representative of the different *Anaplasma* species and *Ehrlichia canis* chosen as an outgroup. The bootstrap consensus tree inferred from 1000 replicates is taken to represent the evolutionary history of the taxa analyzed. Branches corresponding to partitions reproduced in less than 50% bootstrap replicates are collapsed. The percentage of replicate trees in which the associated taxa clustered together are shown next to the branches. This analysis involved 53 nucleotide sequence types, and there were a total of 405 positions in the final dataset. References in clades are: *Ca* A. cinensis [2,12], *A. phagocytophilum*-like1 [17].

**Figure 2 ijms-24-00035-f002:**
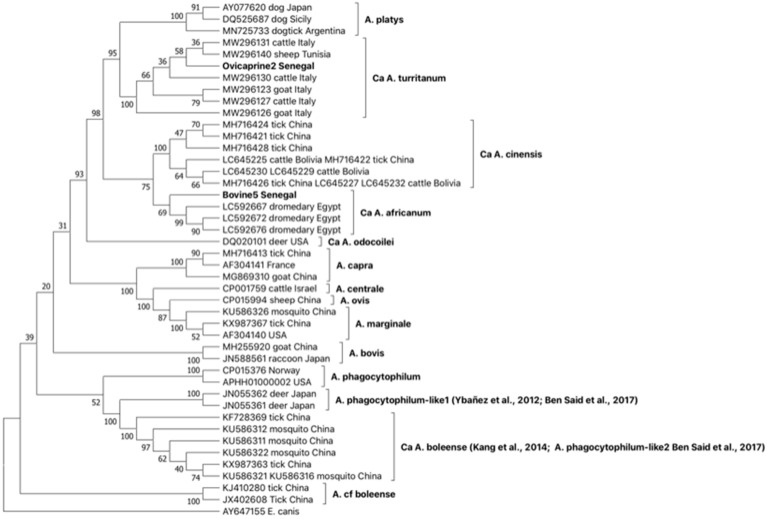
Phylogeny of the 2 *gltA* sequence types identified in Senegal and genetically related to *A. platys* with OTUs selected as representative of the different *Anaplasma* species and *Ehrlichia canis* chosen as an outgroup. The bootstrap consensus tree inferred from 1000 replicates is taken to represent the evolutionary history of the taxa analyzed. The percentage of replicate trees in which the associated taxa clustered together is shown next to the branches. This analysis involved 44 nucleotide sequence types, and there were a total of 483 positions in the final dataset. References in clades are: *A. phagocytophilum*-like1 [17,23], *Ca* A. boleense [20,23].

**Figure 3 ijms-24-00035-f003:**
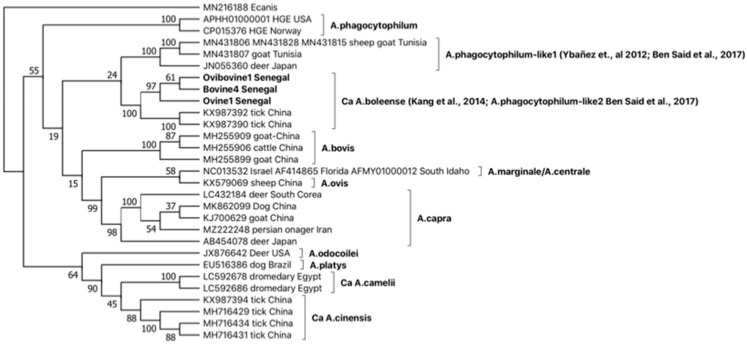
Phylogeny of *groEL* sequence types identified in Senegal genetically related to *A. phagocytophilum* with OTUs selected as representative of the different *Anaplasma* species and *Ehrlichia canis* chosen as an outgroup. The bootstrap consensus tree inferred from 1000 replicates is taken to represent the evolutionary history of the taxa analyzed. The percentage of replicate trees in which the associated taxa clustered together is shown next to the branches. This analysis involved 29 nucleotide sequence types, and there were a total of 667 positions in the final dataset. References in clades are: *A. phagocytophilum*-like1 [17,23], *Ca* A. boleense [20,23].

**Table 1 ijms-24-00035-t001:** Characteristics of Anaplasma species and unclassified genovariants ^1^.

Species	Main Hosts	Pathogenicity	Cell; Tropism	Primary Vectors	Distribution	Reference
*A. phagocytophilum*	Mammals	Vertebrates including human	Granulocytes	*Ixodes* spp.	The Americas, Eurasia, Africa	[1]
*A. platys*	Dog	Cyclic thrombocytopenia in dogs, human infection	Platelets	*Rhipicephalus* spp.	Worldwide	[1,5,6]
*A. bovis*	Cattle	Bovine anaplasmosis, human infection	Monocytes	Various tick species	South Europe, the Americas, Africa, Asia	[1,7]
*A. ovis*	Sheep	Ovine anaplasmosis, human infection	Erythrocytes	*Dermacentor* spp., *Rhipicephalus* spp.	Asia, Africa, Europe, N. America	[8,9]
*A. marginale*	Cattle	Bovine anaplasmosis	Erythrocytes	Various tick species	Worldwide	[1,9]
*A. centrale*	Cattle	Mild anaplasmosis	Erythrocytes	*Rhipicephalus* spp.	Worldwide	[1,9]
*Ca* A. turritanum	ruminants, cats	Not established	Platelets, granulocytes	*Rhipicephalus* spp., *Haemaphysalis* spp.	South Europe, Asia, Africa	[2,10,11]
*Ca* A. cinensis	Cattle	Not established	Not established	*R. microplus*	Asia, Africa	[2,12]
*Ca* A. odocoilei	Deer	Not established	Platelets	*A. americanum*,	USA, Mexico	[13]
*A. capra*	Humans, ruminants, dogs	Human infection	Erythrocytes	Various tick species	Europe, Asia	[14,15]
*A. phagocytophilum* like-1 (japan strains)	Sika deer, cattle, goat, sheep	Not established	Not established	Various tick species	Asia, Europe, Africa	[3,16,17]
*A. phagocytophilum* like-2; (*A. boleense*)	Cattle	Not established	Not established	Various tick species	Asia, Africa	[18,19,20]

^1^ Genovariants based solely on *16S rRNA*, *23S rRNA*, and *rpoB* genes are not considered in this table.

**Table 2 ijms-24-00035-t002:** Positivity of PCR tests specific for *Anaplasma* strains genetically related to *A. phagocytophilum* and *A. platys.*

Host Species	N ^1^	*Ca* A. turritanum				Cf *Ca* A. cinensis		*A. phagocytophilum* Group			
		gltA	groEL	Both gltA; and groEL	At Least 1 Test	gltA	groEL	16S rRNA	groEL; A. Phago	groEL; A. Phago Like 1	groEL; A. Phago Like 2
Sheep	134	81	83	81	83	0	0	23	1	-	18
Goat	28	9	7	6	9	0	0	-	-	-	-
Cattle	14	0	0	0	0	3	3	5	-	-	5
TOT	176	90	90	87	92	3	3	28	1	-	23

^1^ total number of sampled animals.

**Table 3 ijms-24-00035-t003:** Assignment of sequence types obtained in this study and similarity to sequences deposited in the GenBank.

Sequence Type	Gene (Primers Used); Reference	Animal Sources	BlastN	GenBank Accession Number(s)
Ovicaprine1	groEL (EphplgroEL(569)F, EphplgroEL(1193)R, EplgroEL(1084)R); [30]	37 sheep 6 goats	98–100% *Ca* A. turritanum	OP573342-384
Bovine1	groEL (EphplgroEL(569)F, EphplgroEL(1193)R, EplgroEL(1084)R);	1 cattle	96% *Anaplasma* sp. 88% *Ca* A. cinensis	OP573278
Bovine2	groEL (EphplgroEL(569)F, EphplgroEL(1193)R, EplgroEL(1084)R); [30]	1 cattle	96% *Anaplasma* sp. 88% *Ca* A. cinensis	OP573279
Bovine3	groEL (EphplgroEL(569)F, EphplgroEL(1193)R; EplgroEL(1084)R); [30]	1 cattle	96% *Anaplasma* sp. 88% *Ca* A. cinensis	OP573280
Ovine2	groEL (EphplgroEL(569)F, EphplgroEL(1193)R; EphgroEL(1142)R); [30]	1 sheep	88% *A. bovis*	OP573277
Ovibovine1	groEL (APHAGOVAR2GROEL_F, APHAGOVAR2GROEL_R1, APHAGOVAR2GROEL_R2); [3]	3 Cattle 3 Sheep	99% *Anaplasma* sp. 91% *A. boleense*	OP573323, OP573327-28, OP573332-33, OP573338
Ovine1	groEL (APHAGOVAR2GROEL_F, APHAGOVAR2GROEL_R1, APHAGOVAR2GROEL_R2); [3]	13 Sheep	99% *Anaplasma* sp. 91% *A. boleense*	OP573324-26, OP573329-31, OP573334-37, OP573339-41
Bovine4	groEL (APHAGOVAR2GROEL_F, APHAGOVAR2GROEL_R1, APHAGOVAR2GROEL_R2); [3]	1 Cattle	99% *Anaplasma* sp. 91% *A. boleense*	OP573322
Ovicaprine2	gltA (AplaLikeGLTAF1, AplaLikeGLTAR, AplaLikeGLTAF2); [2]	7 goats	99–100% *Ca* A. turritanum	OP573281-321
Bovine5	gltA (Pglt-F, Pglt-R1, Pglt-R2); [12]	3 Cattle	79% *Anaplasma* spp. 78,56% *Ca* A. cinensis	OP654651-53
Ovibovine2	16S rRNA (EE1, EE2, SSAP2f, SSAP2r); [22,23]	4 cattle 15 Sheep	100% *Anaplasma* sp. 99.83% *A. boleense*	OP546293-94, OP546304-05, OP546296-98, OP546301-03, OP546306, OP546309-11, OP546313-16, OP546318
Ovine3	16S rRNA (EE1, EE2, SSAP2f, SSAP2r); [22,23]	2 sheep	99.83% *Anaplasma* sp. 99.66% *A. boleense*	OP546312, OP546317
Ovine4	16S rRNA (EE1, EE2, SSAP2f, SSAP2r); [22,23]	2 sheep	100% *A. phagocytophilum* like2	OP546307, OP546299
Ovine5	16S rRNA (EE1, EE2, SSAP2f, SSAP2r); [22,23]	3 sheep	100% *A. phagocytophilum* like2	OP546300, OP546295, OP546308

## Data Availability

Not applicable.

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
