# Peer review of "Emergence of Anaplasma Species Related to A. phagocytophilum and A. platys in Senegal"

_ijms, 2022, doi:10.3390/ijms24010035_

Round 1

Reviewer 1 Report

Generally, this is a well written and scientifically accurate piece of work. I have only a few comments that I would like the authors to consider/correct.

1) Line 41 - do you mean human disease? be specific.

2) Line 222-224. The difference in prevalence between sheep and goats is described. It would be very simple to use some statistical analysis to demonstrate that the difference is statistically significant (it is!). This should be addressed.

3) A few errors in formatting in the reference section. The scientific names are often written in the (incorrect) following way: Anaplasma Platys. This should be corrected.

Author Response

Reviewer 1

Generally, this is a well written and scientifically accurate piece of work. I have only a few comments that I would like the authors to consider/correct.

We thank the reviewer for his general comments.

1) Line 41 - do you mean human disease? be specific.

According to the reviewer comment we modified this sentence, that now reads: “A. phagocytophilum is the most relevant species in term of animal and human tick-borne diseases…”

2) Line 222-224. The difference in prevalence between sheep and goats is described. It would be very simple to use some statistical analysis to demonstrate that the difference is statistically significant (it is!). This should be addressed.

As we mentioned in the manuscript the number of sampled goats and sheep was not comparable. For this reason, we avoided including statistics, for it could be biased by sampling. We tried to obtain more samples from goats, but as you can understand it is not easy to sample in that regions of Africa.

3) A few errors in formatting in the reference section. The scientific names are often written in the (incorrect) following way: Anaplasma Platys. This should be corrected.

Reference section has been reviewed according to the reviewer comment.

Reviewer 2 Report

Comments to the Authors:

The paper describes a study on the presence of Anaplasma strains related to the zoonotic A. phagocytophilum and A. platys in Senegal ruminants by combining molecular tools targeting the Anaplasma 16S rRNA, groEL and gltA genes. For this, a total of 176 EDTA blood collected from clinically healthy ruminants, sheep (n=134), goats (n=28), and cattle (n=14), originating from Keur Momar Sarr Senegal region, were tested. The findings confirm that Anaplasma species diversity is greater than expected and implications of the emergence of Anaplasma species in Sub Saharan Africa on diagnostics, diversity, transmission, and public health are also discussed.

The question posed by the authors is, therefore of interest. However, some changes should be corrected before accepting for publishing, as following.

 The Abstract – Overall, the Abstract include too many general statements, instead of basic information about the study design and findings. Actually, the authors describe their findings in the aim of the study (page 4, lines 76-82) , which should be only about the aim. Therefore, it is suggested to describe better the findings within the abstract and to re-describe the aim, accordingly.

Line 15:  ……”Anaplasma” should be Italicized <Anaplasma>

Line 18: ….”Although species related to the zoonotic A. platys and A. phagocytophilum have been reported in several African and European Mediterranean-facing Countries..” – ..I think a better phrase could be used, as “…–facing countries”  is not a common expression .

Line 21:  “ Results demonstrated a recent introduction of Ca A. turritanum” –  The strain is mentioned first time in the Abstract, therefore, a complete name should be provided, for the benefits of all readers, then the abbreviate name. Also, a short description should be provided, instead of just mention it – i.e.  it is a species related to A. platys / A. phagocytophilum – in order to be consistent with the title and the aim of the study.

1. Introduction

Line 56: same suggestion for “.. Ca Anaplasma boleense ..” >> Candidatus (Ca) Anaplasma boleense

Lines 74 – 82: the paragraph is supposes to describe the aim of the study. However, both aim of the study, findings and conclusion statement are all combined. Therefore, it is suggested to keep only the aim, and the rests to be provided in the appropriate section.

 2. Results

Line 132: and also in the Materials and methods: the authors use “…heminested PCR”, instead of the more commonly used Could authors explain or comment on it, in order to avoid any potential confusion?

3. Discussion

Line 211: it is also, a quite strange expression about “ .. Beside the canonical 6 species included”, could be better used . the widely recognized six..>

Line 217: ..the same suggestion, as above, with regards to the expression “several European and African Mediterranean-facing Countries ...

Line 219: “This manuscript” – it sounds better <..this study>.

Lines 221 - 223: there is a paragraph not well comprehensible. ”A. turritanum was detected in some 50% of tested ruminants from Senegal, with a prevalence consistent with values observed in the Mediterranean area [10]. Ca A. turritanum  prevalence seems higher in sheep (83/134, 62%) respect to goats (9/28, 32%), although..”

Line 246: similarly, “Ca A. africanum clade includes strains previously rescued in dromedary camels”

Line 247: “..Anaplasma” – Italicized <Anaplasma>

Lines 261- 264: the concluding paragraph includes general statements, highlighting the relevance of the findings. However, some of the recommendations would not be feasible (262-264), but could be of discussion for the future research.

 4. Materials and Methods

Line 275: “ 176 clinically healthy ruminants ”; it would of readers’ benefits to add here some info about animals, i.e. sheep (n=134), goats (n=28), and cattle (n=14).

 References

Line 367, 378, 419,  : the year – should be Bold

Lines 284 – 295: please see the comments above, about the use of “heminested PCR”, if corrections is to be done, to have into consideration.

Author Response

Reviewer 2

The paper describes a study on the presence of Anaplasma strains related to the zoonotic A. phagocytophilum and A. platys in Senegal ruminants by combining molecular tools targeting the Anaplasma 16S rRNA, groEL and gltA genes. For this, a total of 176 EDTA blood collected from clinically healthy ruminants, sheep (n=134), goats (n=28), and cattle (n=14), originating from Keur Momar Sarr Senegal region, were tested. The findings confirm that Anaplasma species diversity is greater than expected and implications of the emergence of Anaplasma species in Sub Saharan Africa on diagnostics, diversity, transmission, and public health are also discussed.

The question posed by the authors is, therefore of interest. However, some changes should be corrected before accepting for publishing, as following.

We thank the reviewer for this general comment. Below, we provide a point by point detail answer.

 1 Abstract

 Overall, the Abstract include too many general statements, instead of basic information about the study design and findings. Actually, the authors describe their findings in the aim of the study (page 4, lines 76-82), which should be only about the aim. Therefore, it is suggested to describe better the findings within the abstract and to re-describe the aim, accordingly.

According to the reviewer, abstract has been modified as follow:”

The genus Anaplasma (Anaplasmataceae, Rickettsiales) includes tick-transmitted bacterial species of both Veterinary and Human medicine importance. Beside the traditionally recognised 6 Ana-plasma species (A. phagocytophilum, A. platys, A. bovis, A. ovis, A. centrale, A. marginale) novel strains and candidate species, of relevance to veterinary and human medicine are emerging worldwide. Although species related to the zoonotic A. platys and A. phagocytophilum have been reported in several African and European Mediterranean Countries data on the presence of these species in Sub-Saharan Countries are still lacking. This manuscript reports the investigation of Anaplasma strains related to zoonotic species in ruminants in Senegal, by combining different molecular test and phylogenetic approaches. Results demonstrate a recent introduction of Candi-datus (Ca) Anaplasma turritanum, a species related to the pathogenic A. platys, possibly originat-ing by founder effect. Also, novel undetected strains related to Candidatus (Ca) Anaplasma cinen-sis are detected in cattle. Based on groEL and gltA molecular comparisons, we propose including these latter strains into the Candidatus (Ca) A. africanum species. Finally, we also report the emer-gence of Candidatus (Ca) A. boleense in Senegal. Collectively, results confirm that Anaplasma spe-cies diversity is greater than expected and should be further investigated, and that Anaplasma routine diagnostic procedures and epidemiological surveillance should take into account speci-ficity issues raised by the presence of these novel strains, suggesting the use of a One Health ap-proach for the management of Anaplasmataceae in Sub Saharan Africa.”

  1. Introduction

Line 56: same suggestion for “.. Ca Anaplasma boleense ..” >> Candidatus (Ca) Anaplasma boleense

Done

Lines 74 – 82: the paragraph is supposes to describe the aim of the study. However, both aim of the study, findings and conclusion statement are all combined. Therefore, it is suggested to keep only the aim, and the rests to be provided in the appropriate section.

We understand this reviewer suggestion. However, we believe that anticipating results help the reader in following the result section and discussion. Also, the first reviewer did not make this type of suggestion. We would therefore keep this section as it was, if this is not compulsory.

  1. Results

Line 132: and also in the Materials and methods: the authors use “…heminested PCR”, instead of the more commonly used Could authors explain or comment on it, in order to avoid any potential confusion?

Now we use semi-nested throughout the text

  1. Discussion

Line 211: it is also, a quite strange expression about “ .. Beside the canonical 6 species included”, could be better used

This sentence has been modified according to the reviewer comment

Line 217: ..the same suggestion, as above, with regards to the expression “several European and African Mediterranean-facing Countries ...”

We now avoid the use of “facing in the new version”

Line 219: “This manuscript” – it sounds better <..this study>.

done

Lines 221 - 223: there is a paragraph not well comprehensible.

This paragraph was modified into:” A. turritanum was detected in some 50% of tested ruminants from Senegal, with a prevalence consistent with values previously reported in the Mediterranean area”

Line 246: similarly, “Ca A. africanum clade includes strains previously rescued in dromedary camels”

According to this comment sentence now reads “Ca A. africanum clade includes strains previously described in dromedary camels from Egypt clade….”

Line 247: “..Anaplasma” – Italicized <Anaplasma>

done

Lines 261- 264: the concluding paragraph includes general statements, highlighting the relevance of the findings. However, some of the recommendations would not be feasible (262-264), but could be of discussion for the future research.

We would keep this sentence as it is, if possible.

  1. Materials and Methods

Line 275: “ 176 clinically healthy ruminants ”; it would of readers’ benefits to add here some info about animals, i.e. sheep (n=134), goats (n=28), and cattle (n=14).

As this numbers are reported in the table with would not use space in the text.

 References

Line 367, 378, 419,  : the year – should be Bold

done

Lines 284 – 295: please see the comments above, about the use of “heminested PCR”, if corrections is to be done, to have into consideration.

done